# Constructing a Flood-Adaptive Ecological Security Pattern from the Perspective of Ecological Resilience: A Case Study of the Main Urban Area in Wuhan

**DOI:** 10.3390/ijerph20010385

**Published:** 2022-12-26

**Authors:** Hongyi Chen, Yanzhong Liu, Lin Hu, Zuo Zhang, Yong Chen, Yuchuan Tan, Yufei Han

**Affiliations:** 1College of Resource and Environmental Engineering, Wuhan University of Science and Technology, Wuhan 430081, China; 2School of Public Administration, Central China Normal University, Wuhan 430070, China

**Keywords:** SCS-CN model, flood adaptation, ecological security pattern, resilient city

## Abstract

The frequent occurrence of floods in urban areas caused by climate change challenges urban resilience. This research aims to construct an ecological security pattern (ESP) that is adaptive to floods to enhance urban resilience in the hope that it will help cities cope with floods better. In this research, the main urban area of Wuhan (WUH) represents the study area. The lakes were selected as the ecological sources and the Soil Conservation Service-Curve Number (SCS-CN) model was used to calculate the runoff volume corresponding to each land type and, based on this, assign resistance values to the land types; as such, the land type surface is referred to as the runoff resistance surface, and the runoff resistance surface is then modified by ecosystem service capabilities. The Minimum Cumulative Resistance (MCR) model was used to extract the connecting corridors between the sources. This research plan includes 18 ecological sources, 10 key ecological corridors, and 22 potential ecological corridors, with a total length of about 344.21 km. Finally, it provides a *two-axis* and three-core urban ecological resilience optimization strategy for decision makers and a new approach for controlling floods in urban areas from the perspective of ecological resilience.

## 1. Introduction

Due to climate change and urbanization, floods in urban areas are occurring more frequently. Climate change also causes extreme rainfall [1,2], while urbanization increases impervious underlying surfaces, causing waterlogging [3,4,5]. The traditional flood control and drainage measures promote the construction of traditional gray infrastructure, such as drainage pipes and flood control revetments. Although these facilities are flood-resistant, their static defense cannot fully adapt to the increasing flood risks and cannot automatically recover after damage occurs. Therefore, gray infrastructure is not sustainable. In other words, cities should develop more effective ecological methods to prevent floods [6].

As a comprehensive urban development concept, a *resilient city* is characterized by its flexibility, adaptability, and transformative ability [7]. Likewise, it provides theoretical support for solving current flood-related problems in cities. Many cities have introduced resilient retrofit programs, such as New York [8], Melbourne [9], Rotterdam [10], etc. In China, a sponge city represents a comprehensive solution for various water-related problems, such as water shortage, water pollution, and floods [11]. Researchers conducted extensive studies on the planning and design [12,13] and index control [14], including the unit area, runoff coefficient, expected rainfall, etc. There has also been significant exploration of the system construction [15] of sponge cities, but researchers have also conducted case studies [16] on some pilot areas in sponge cities, including the synergy between green and gray infrastructure [17,18], flood control capability [19], contamination risks [20], etc. Some studies also optimize the layout of sponge cities in terms of the rainfall runoff, pollutant discharge, minimization of construction and operation costs, and maximization of environmental benefits [21]. A sponge city improves the storage capacity of the underlying surface, regulates the infiltration, interception, storage, purification, reuse, and discharge of rainwater, and enhances flood resilience of the city through disaster reduction. However, sponge cities have exposed many deficiencies in the process of their construction and use. These deficiencies are the ecological risks to soil and groundwater caused by the decomposition of the Sponge City filler during rainfall, cumulative effect of pollutants, ecological pressure and economic burden caused by the clogging of the filling and the recycling of the waste fill, weak ecological service capability, uneven distribution, and lack of integrity [6]. This means that cities should be made more resilient from an ecological perspective.

While some studies [22,23,24] have systematically evaluated the urban resilience of Wuhan city and proposed detailed guidance schemes for policy makers, this study will be conducted from the perspective of spatial analysis and ecological planning. The resilient city theory holds that urban ecosystems with complex structures resist disturbances from outside the city better. The ecological security pattern (ESP) is a spatial configuration scheme that combines the ecological or human elements in the region through a combination of points, lines, and surfaces [25], and is a complex network structure with a holistic and multi-patch [26] connectivity designed to adapt to floods, complement the sponge city, and enhance urban ecological resilience. Most traditional research on ecological security patterns follows a single ecological protection perspective [27,28,29] or combines human activities [30,31], natural geographical elements (such as slope and elevation) [32,33], habitat protection [34], economy [35,36], etc. However, the study of the interaction between flooding and ecological security patterns is not well understood. Some studies [26] have also proposed ecological improvement measures for the central area of Wuhan from the perspective of spatial analysis, but their results cannot effectively guide the construction of ecological corridors within the urban area.

This research takes the main urban area of Wuhan (WUH) as the study area in order to enhance the ecological resilience of the city and study the flood adaptability of the urban ESP. The lakes are protected as ecological sources, the blueways and greenways are combined with each other as ecological corridors which look like river parks, and ecological corridors are planned to connect the waters of the lakes to each other to form a holistic and resilient ecological network. In this, the blueways can quickly divert runoff during heavy rainfall and the interconnected lakes have greater storage capacity, which can increase the threshold of precipitation needed for creating urban flooding. In addition, to differentiate from the traditional gray infrastructure, the study incorporates ecosystem service functions to ensure that the ESP provides ecological services to the city even during non-flooding periods. This study also introduces a gravity model to guide the priority of corridor construction and field investigations to find a target paradigm for corridor construction, which hopefully will help relevant authorities to take better relevant measures. The main urban area of WUH is key for rainwater management and control. Research into the ESP that is adaptive to floods can provide theoretical and methodological support for WUH and other cities to improve their ecological resilience.

## 2. Study Area

The main urban area of WUH is affiliated with Wuhan City in the Hubei Province. It covers an area of 450 km^2^ [37] and is located at the intersection of the Yangtze and Hanjiang Rivers. Wuhan has many lakes, namely the East Lake, Yanxi Lake, South Lake, Moon Lake, MoShui Lake, Nantaizi Lake, Wanjia Lake, etc. Due to its location in the alluvial plain in the middle reaches of the Yangtze River, the soils in the study area are dominated by paddy and chao soil [38] under the influence of groundwater movement and farming activities, both of which are characterized by a clay-like texture and poor permeability. Wuhan’s climate is humid and subtropical, characterized by monsoons, with an annual average temperature of 17.4 °C, a mean monthly maximum temperature of 30.8 °C, an annual average rainfall of 129 days, and an annual average rainfall of 1317.5 mm (the 2009–2019 average) [39]. In 2021 [40], the average precipitation in May to October during the flood season was 998.2 mm, accounting for 74.3% of the annual precipitation; the average precipitation from mid-June to July was 329.5 mm, accounting for 24.5% of the annual precipitation; the month with the most precipitation was August, with 310.0 mm; the least precipitative month was December, with 8.7 mm. There are 574 million m^3^ of surface water resources in the main city of Wuhan, with 791.2 billion m^3^ of transit water.

In addition to the Yangtze River and Han River, there are three major water systems: the Daoshui River, the Nieshui River, and the Jushui River. There are also many tributary systems, such as the Tongshun River and the Jinshui River [41]. The river network is crisscrossed and the water system is huge. Wuhan is known as the *City of a Hundred Lakes* and the *city of wetlands*, and the lake ecosystem plays a decisive role in urban flood control. Flood control has been very effective in WUH, but there are still certain risks, mainly with three aspects: (1) there are many water systems and the pressure on flood control is high; (2) extreme weather is increasing, so the risk of severe floods is still present; (3) waterlogging is frequent. In addition, urban construction has encroached upon many lake ecosystems [42], which increased Wuhan’s vulnerability to severe floods. According to *People’s Daily* [43], the lakes in Wuhan have shrunk by 106 km^2^ in the past three decades. These shrinking lakes continue to weaken the ecological service capacity, threatening regional ecological security and greatly weakening urban ecological resilience. The location and land use types of the study area are shown in Figure 1.

In the map above, the Hubei Province is in south-central China. The location of the main urban area of Wuhan within the city is shown in the upper right part of the box.

## 3. Methodology and Data Sources

### 3.1. Data Sources

The daily data of precipitation come from Hubei Meteorological Administration, the elevation data (DEM) with 30 M accuracy from the Chinese Academy of Sciences Geospatial Data Cloud (http://www.gscloud.cn/search, accessed on 18 November 2021), and the land use data in the main urban area of Wuhan come from the Wuhan Bureau of Natural Resources and Planning (ghttp://zrzyhgh.wuhan.gov.cn/zwgk_18/fdzdgk/ghjh/zzqgh/202001/t20200107_602858.shtml, accessed on 20 November 2021), Wuhan soil data with 30 M accuracy comes from the Harmonized World Soil Database (https://www.fao.org/soils-portal/soil-survey/soil-maps-and-databases/harmonized-world-soil-database-v12/en/, accessed on 10 December 2021), the spatial distribution data of waterlogging points are from the Wuhan Municipal Water Affairs Authority (http://swj.wuhan.gov.cn/tzdt/jcss/202005/t20200511_1309387.html, accessed on 25 December 2021), the net primary productivity (NPP) data of vegetation in the main urban area of Wuhan with 30 M accuracy comes from the Resource and Environmental Science and Data Center of the Chinese Academy of Sciences (https://www.resdc.cn/, accessed on 10 March 2022), the data of points of interest (POI) in the main urban area of Wuhan are from Baidu Map Open Platform (https://lbsyun.baidu.com/, accessed on 25 December 2021), and the rest of the meteorological data in Wuhan comes from the “*Wuhan Statistical Yearbook*” (2010–2020) (http://tjj.wuhan.gov.cn/tjfw/tjnj/, accessed on 18 November 2021).

### 3.2. The Methodological Framework

Sources are patches that influence the stability, integrity, and functionality of ESP and are the basis of ESP. In this research, lakes in the main urban area of WUH were selected as ecological sources. The Soil Conservation Service-Curve Number (SCS-CN) model is used to calculate the surface runoff volume of each category after a rainstorm, which is spatially expressed in ArcGIS and transferred to a raster map after assigning resistance values, called the runoff resistance surface. Likewise, the water connotation function and the recreation and leisure function are rasterized and assigned resistance values, which are called water connotation resistance surface and recreation and leisure resistance surface. The raster calculator tool of ArcGIS is used to superimpose the three resistance surfaces to get the comprehensive resistance surface (the quantitative expression of the degree of difficulty of moving matter and energy through a certain surface). Ecological corridors were constructed using the Minimum Cumulative Resistance (MCR) model. Furthermore, the location of ecological corridors was optimized based on the runoff path (Appendix A) and the waterlogging points (Appendix A) issued by the Wuhan Municipal Water Affairs Authority. Finally, the gravity model was used to prioritize the construction of the ecological corridor. The specific experimental flow is shown in Figure 2.

### 3.3. Runoff Calculation Based on the SCS-CN Model

The variety of land types have their own impervious areas and unique substrate properties, which determine their different runoff-generating capacities in the face of rainfall processes. In this research, the SCS-CN model [44] was used to calculate the runoff volume generated by different land types after heavy rain, and then the values of the runoff volume were connected with the vector land type distribution map in ArcGIS to derive the spatial distribution of runoff volume corresponding to different land types. The SCS-CN model is a simple model with a clear physical concept and few required parameters [45]. The regular formula for the SCS-CN model is [46]:(1)$$\frac{F}{S}=\frac{Q}{P-I_{a}}$$
where *P* represents the total precipitation (mm), and $I_{a}$ the initial abstraction (mm), including the initial loss of ground filling, interception, surface water storage, and infiltration. *F* is the loss after the generation of surface runoff, i.e., the actual cumulative infiltration (mm) (excluding $I_{a}$); *Q* is the direct runoff (mm); *S* is the possible retention amount (mm).

The water balance equation is as follows [47]:(2)$$P=I_{a}+F+Q$$

Combining Equation (1) with Equation (2), eliminating *F*, we obtain Equation (3) [48]. It should be noted that runoff cannot be generated when the initial loss is satisfied.
(3)$$\{\begin{array}{c} Q=\frac{{\left(P\;-\;I_{a}\right)}^{2}}{P\;+\;S\;-\;I_{a}},\;P\geq 1\\ Q=0,\;P<1 \end{array}$$

Equation (3) is the equation for calculating the yield flow of the SCS-CN model. Since $I_{a}$ data were not obtained, the parameter initial loss rate *λ* was used to establish a linear relationship between $I_{a}$ and S [49]:(4)$$I_{a}=\lambda S$$
where *λ* usually has the standard value of 0.2 [44]. If the value of *λ* is 0.2, the yield calculation formula of the surface runoff of the SCS-CN model is [49]:(5)$$\{\begin{array}{c} Q=\frac{{\left(P\;-\;0.2S\right)}^{2}}{P\;+\;0.8S},\;P\geq 0.2S\\ Q=0,\;P<0.2S \end{array}$$

The *S* value in all of the above equations is usually calculated from the equation with the CN value parameter [49]:(6)S=25,400/CN−254(international system of units)

In the formula, CN is a dimensionless constant reflecting the antecedent soil moisture condition (AMC) in the basin, and CN values are influenced by multiple factors such as land types, vegetation, and soil texture [46]. Since the United States and China have different land types, the original CN values could not be used directly in this study, and the optimization and adaptation methods of the model are not focused on here. This study also concluded that, in urban areas, CN values are more likely to be influenced by land types, so this research used the CN value of Suzhou City [50], which has similar land types, soil textures, and climatic and hydrological conditions to Wuhan City, as a reference to correct the CN value. According to the soil texture and the model standards, soil is divided into four categories: A, B, C, and D (Table 1) [46]. The content of clay is relatively high in Wuhan, making it a part of category C. Therefore, the CN value under category C was selected. AMC was divided into three types: AMC-I (dry), AMC-II (general), and AMC-III (wet), according to precipitation in the previous five days. Based on historical records of rainfall in Wuhan, this research selected the rainfall value of the past ten years starting from 6 July 2016, which marked a 12 h long rainfall of 179.3 mm in a single day. The precipitation in the first five days was moderate, so AMC-II conditions were selected to simulate the runoff in the study area.

Since the generation of flood risk does not depend entirely on the runoff volume (runoff depth) but also on the distribution area of each land type in the study area, in order to investigate the contribution of different land types to the volume of rainwater retained on the surface of the study area, this study used ArcGIS to count the distribution area of different land types in the study area and calculate the volume of runoff corresponding to different land types. In this study, the total runoff volume was used as a proxy for the runoff volume.

### 3.4. Water Conservation Capability Assessment

Water conservation capacity is the ability of an ecosystem to regulate water flow and the water cycle [51]. The overflow of sediments in the Yangtze River and the urban expansion of Wuhan have affected lakes and wetlands, weakening their water conservation capacity. Water conservation services function by enhancing soil infiltration and moderating surface runoff [52] to improve the ecosystem service capacity of the ESP while correcting the resistance distribution of runoff. This research used the water conservation capacity to modify the simulation results of the SCS-CN model. The ecosystem water conservation service capability index [53] was selected for the evaluation index of water conservation. The formula is as follows [53]:(7)$$WR=NPP_{mean}\times F_{sic}\times F_{pre}\times \left(1-F_{slo}\right)$$
where *WR* is the ecosystem water conservation service capacity index; $NPP_{mean}$ is the annual average net primary productivity of the ecosystem; $F_{sic}$ is the soil infiltration capacity factor; $F_{pre}$ is the annual average precipitation; $F_{slo}$ is the slope value. All four values were normalized. Finally, the four factors were calculated by overlaying them using ArcGIS’s raster calculator tool.

### 3.5. Leisure and Recreation Capability Assessment

In addition to flood control and ecological functions, leisure and recreational functions are also important features that distinguish ESP from traditional gray infrastructure. The deeply buried pipeline has a single function and does not provide an additional service during non-flooding periods, while the ecological corridor combining the blueway and greenway not only has better flood resistance during flooding periods, but also provides leisure and recreational services for urban residents during non-flooding periods. The accessibility of ecological space and cultural services in residential areas is important for promoting the physical and mental wellbeing of residents [54]. This study uses a crawler tool called easypoi to crawl the point data coded as parks, scenic spots, and green spaces in the *Baidu map open platform*, generate vector point data, and import them into ArcGIS. With the Kernel Density tool in ArcGIS, the density of the POI data of the ecological space and cultural services in the main urban area of Wuhan was calculated and used as the basis for evaluating the accessibility of the ecological space.

### 3.6. The MCR Model

In order to find the corridor development and construction mode with the lowest impact and the best drainage effect, it was necessary to extract the lowest cost path between the various sources. Based on the MCR model, this research used ArcGIS to extract the lowest cost path between each ecological source:(8)$$MCR=\int_{min}\sum \left(D_{ij}\times R_{i}\right)$$
where $D_{ij}$ refers to the field distance from lake *j* to the environmental unit *i* in the region; $R_{i}$ refers to the resistance coefficient of environmental unit *I*; $R_{i}$ represents the ease of passage of matter and energy [55].

### 3.7. DEM-Based Runoff Path Analysis

Runoff paths are the locations of the most likely stream channels in the study area extracted from DEM data using ArcGIS hydrologic analysis, and are contiguous segments of a number of continuous low elevation locations where rainfall is more likely to collect in the study area during rainfall. The hydrological analysis tool was used to extract the stream channels and compare the maps to remove the existing streams to obtain the potential stream channels in the study area. These potential channels can be used as a basis for guiding the construction of artificial rivers. According to the spatial distribution of runoff paths, the optimization of the urban ESP enhances the infiltration capacity of ecological corridors and maximizes ecological benefits, which is in line with the requirements of urban elastic transformation.

### 3.8. Gravity Model

The development and construction of ecological corridors are usually carried out in stages. In order to provide guidance on timing the construction of corridors, the gravity model was used to calculate the strength of the interaction between ecological sources. The data were then used to judge the relative importance of each corridor. The formula is as follows [56]:(9)$$G_{ij}=\frac{N_{i}N_{j}}{D_{ij}{}^{2}}=\frac{\left[\frac{1}{P_{i}}\times ln\left(S_{i}\right)\right]\left[\frac{1}{P_{j}}\times ln\left(S_{j}\right)\right]}{{\left(\frac{L_{ij}}{L_{max}}\right)}^{2}}=\frac{L_{max}^{2}lnS_{i}lnS_{j}}{L_{ij}^{2}P_{i}P_{j}}$$
where $G_{ij}$ is the interaction force between sources *i* and *j*; $N_{i}$ and $N_{j}$ are the weighting values of sources *i* and *j*, respectively; $D_{ij}$ is the normalized value of the minimum cumulative resistance between sources *i* and *j*; $P_{i}$ and $P_{j}$ are the resistance values (extracted from the raster layer attribute table of the ecological corridor) of sources *i* and *j*; $S_{i}$ and $S_{j}$ are the areas of sources *i* and *j*; $L_{ij}$ is the minimum cumulative resistance value between sources *i* and *j*; $L_{max}$ refers to the maximum cumulative resistance of all the corridors in the study area.

### 3.9. ESP Construction Principles

#### 3.9.1. Selection of Ecological Sources

The ecological source is the source of species dispersal, the place where rainwater collects, and the cornerstone of urban resilience. It should have the following characteristics: ① a higher ecosystem service value; ② a larger water volume; ③ a larger land area. A higher ecosystem service value can ensure a stable provision of ecological services and a higher protection value [57]. A larger water volume can reduce the flood peaks and regulate and store water from floods. A larger land area can ensure the stability of the ESP.

#### 3.9.2. Construction of Resistance Surfaces

The ESP that is adaptive to floods should integrate the three functions of flood regulation and storage and ecological and cultural services. Therefore, this research selected the spatial distribution of the runoff simulated in the SCS-CN model as the basic resistance surface. Furthermore, it used the spatial distribution of water conservation and recreation capacity to correct the resistance surface. Because the absolute value of resistance has no practical significance, and since the tendency to assign resistance is more important [58], this research chose 1–50 as the assignment interval and assigned runoff resistance values to different patches based on the degree of development impact, construction cost, and stormwater management and control goals. The difficulty of constructing and developing ecological corridors and the cost of construction were proportional to the resistance value. The lower the construction difficulty and cost, the lower the resistance value. To regulate rainwater, this research used the runoff as the basis for resistance assignment. The larger the total runoff volume was, the lower the resistance value of the corresponding land type was. The next step was to connect the resistance values to the land type and then convert to raster output. The two ecosystem service capacity values are then used as the basis for assigning resistance values. These three resistance surfaces are combined to obtain the integrated resistance surface.

#### 3.9.3. Extraction of Ecological Corridors

Based on the MCR model, the shortest path from one source to the other was extracted as the base ecological corridor using the cost distance and cost path analysis tools of the distance analysis module in ArcGIS. Through hydrological analysis, the runoff path data were extracted from DEM. Based on the waterlogging risk points in the main urban area, the coordinates of each waterlogging road section were imported into ArcGIS. Using the waterlogging risk points as the source points, the ecological corridors between the waterlogging risk points were obtained through the MCR model. The ecological corridor between the runoff path and each waterlogging risk point was used to optimize base ecological corridors.

## 4. Results

### 4.1. Corresponding Runoff Analysis by Land Type

ArcGIS was used to vectorize the land use planning map of the main urban area of Wuhan (Figure 3) and count the area corresponding to each land type. The total runoff volume corresponding to each land type is shown in Table 2 and its distribution is shown in Figure 4.

According to Figure 4, bodies of water, roads, industrial land, and residential areas are the most likely to collect rainwater. In particular, roads had the largest CN value and area, generating the most surface runoff. This suggests that stormwater management and road control should be priorities. The main reason for the great influx of industrial real-estate is the area’s considerable size. The area also has a rigid underlying surface with a high CN value, resulting in a high capacity for rainwater retention. Residential areas and educational land generated more surface runoff as they accounted for a quarter of the total study area.

### 4.2. Distribution Characteristics of Two Ecosystem Services

The spatial distribution of the two ecosystem services showed a complementary trend (Figure 5 and Figure 6). In terms of their water conservation capacity, high value areas were mainly distributed in ① the East Lake–Yanxi Lake area, Hanyang Ecological Zone (an area with a well-developed internal structure and a complete ecological function that can exist on its own.), ② the Zhujia River–Fu River area, ③ the Wuchang Institute of Technology–Hubei University of Technology, and ④ the East of Hanyang Ecological Zone, while the low value area was located in the Hankou and Wuchang Ecological Zones. When it comes to their leisure and recreation capacity, the high value areas were mainly concentrated in ⑤ the South of Hankou Ecological Zone, ⑥ the Wuhan Garden Expo Park, ⑦ the Moon Lake–Moshui Lake area, ⑧ the intersection between the Yangtze and the Hanjiang Rivers, and ⑨ the East Lake Scenic Area. In terms of the accessibility of ecological spaces and cultural services, residents of urban centers have an advantage over those in the urban periphery.

### 4.3. Construction of a Flood-Adaptive ESP

#### 4.3.1. Distribution of Ecological Sources

According to the geographical and hydrological characteristics of Wuhan and its urban land type, 18 lakes in the main urban area were selected as ecological sources (Figure 7). It was found that the total area of the ecological sources is 29.54 km^2^, accounting for 6.56% of the study area. As for mountainous and woodland areas, both were excluded from this paper because of their small size and weak flood storage capacity. 

#### 4.3.2. Construction of the Resistance Surface

The distribution of runoff resistance values is shown in Figure 8c. Low resistance values were mainly concentrated on roads and bodies of water. Compared to residential, commercial, and financial land types, the construction and development of roads and bodies of water had the advantages of a lower development impact and low costs.

The values of water conservation and leisure and recreation capacity are negatively correlated with the resistance values. The distribution of the resistance values for leisure and recreation and water conservation are shown in Figure 8a and b, respectively. The three raster resistance surfaces a, b, and c were overlaid using the Raster Calculator tool in ArcGIS. The comprehensive resistance is shown in Figure 8d. After the correction, the comprehensive resistance was found to be low in the center and high in the surrounding area. The low value was mainly distributed in ① the Yiyuan Community–Chezhanlufuren Community and ② the Changfeng sub-district–Garden Expo Park sub-district in the Hankou Ecological Zone, ③ Parrot sub-district–Wulidun sub-district and ④ Xianzheng Street Community in the Hanyang Ecological Zone, ⑤ the Xinhe sub-district–Jiyuqiao sub-district–Zhonghua Road sub-district, and ⑥ the East Lake–Yangchun Lake Community–Beiyangqiao Community area in the Wuchang Ecological Zone.

#### 4.3.3. Identification and Optimization of Corridors

The distribution of ecological corridors is shown in Figure 9.

As shown in Figure 9, there are 19 ecological sources (As shown by the bolded numbers) and 32 ecological corridors (As shown by the circled numbers) in the study area, 9 of which are in the Hankou Ecological Zone; they are generally scattered and long. Of these nine corridors, Corridors No. 6 and 9 meet on the west side of Wuhan No. 17 Middle School. In total, 16 ecological corridors were found in the Hanyang Ecological Zone, accounting for half of the total of ecological corridors in the study area. Their distribution is concentrated, and the corridors are generally short, of which Corridors No. 10, 6, and 9 intersect on the west side of Wuhan No. 17 Middle School. Seven ecological corridors were found in the Wuchang Ecological Zone, roughly showing the spatial distribution pattern of *three verticals and three horizontals,* of which Corridors No. 26 and 30 are connected to the north side of Hubei Shuiguohu High School. Corridors No. 29 and 28 meet at Shouyi Park, while Corridors No. 29 and 32 meet south of the Wuchang Institute of Technology. The specific distribution of these 32 corridors is shown in Appendix A.

According to Figure 9, the ecological corridors in the Wuchang Ecological Zone have the most extensive connectivity range. Although there are fewer communication channels between the ecological sources, the source area is relatively large. Furthermore, the ecological corridors in the Hanyang Ecological Zone are small in scale but strong in connectivity, and the corridors are compact and short. The ecological source area of the Hankou Ecological Zone is small and scattered, resulting in long corridors. Therefore, the width of the corridors should be increased to enhance their anti-interference ability and ensure that the ecological corridors function properly.

### 4.4. Identification of Construction Priorities

Based on the gravity model, this research constructed a matrix of interactions between ecological sources (Appendix A) to identify the strength of the interaction between ecological sources and select key ecological corridors (Figure 10).

The key corridors identified by the gravity model should become priorities for construction so as to reduce the impact of corridor development and improve the efficiency of corridor construction. From Figure 9 and Appendix A, it can be seen that the strength of the connection between source 2 (Chestnut Lake–Huanzi Lake Group) and 5 (Moon Lake) was weak. However, Corridor 7, as a key link between the north and south of the Hanjiang River, was critical for the overall connectivity of the study area and should be given priority. The strength of the connection between source 11 (Tang Lake) and 13 (South Taizi Lake) was also weak. Corridors No. 24 and 25 were important passages connecting Tanghu Lake and the Hanyang Ecological Zone. They were used as key corridors for construction, which can improve the ecological service capacity of the southern part of the Hanyang Ecological Zone. Overall, the strength of the connection between the sources weakened from west to east and from south to north. In particular, the strength of the connection between Tazi Lake and other sources in the northwest was the weakest.

By ArcGIS statistics, the total cumulative length of the 32 ecological corridors in the main urban area of Wuhan is 344.21 km. In total, 10 key corridors with a total length of 63.56 km were mainly distributed around the East Lake and the Hanyang Ecological Zone. There were 22 potential corridors with a total length of 280.65 km located north of the Hanjiang River and south of Wuchang District. By planning ecological corridors to connect to important ecological sources north and south of the main urban area and adding the ecological resources of the Yangtze River to the overall cycle of the ESP, the whole study area can respond to the floods in a more dynamic, proactive, and resilient manner.

## 5. Discussion

### 5.1. Comparison of Current Status

In the past decade or so, the Wuhan municipal government has proposed a plan to improve the ecology of the region, called *Six Lakes Linkage* [59], which would connect some of the rivers and lakes through canals. Therefore, this paper compares the results of the study with the current situation and investigates the merits and shortcomings of the plan.

Firstly, comparison of real-time satellite maps shows that corridors already exist between the four ecological sources (source 5, 6, 7, and 8). Four existing corridors were found in the right location as the research results, namely, Corridors No. 12, 13, 14, and 15. There were two corridors with partially identical results, i.e., the North Prince Lake–Haitang section of Corridor No. 19 and the East Lake–Shahu section of Corridor No. 26.

Secondly, two existing corridors were mislocated. In particular, Corridor No. 2 was mislocated and not long enough. Due to the construction difficulty, the Begonia Road–South Taizi Lake section of Corridor No. 19 cut a corner and took a straight route, abandoning the closer route along Jiangcheng Avenue and opting for one block farther along Wisteria Road. There was already a corridor (East Lake Harbor) connecting the Yangtze River to Yangchun and East Lake. However, the curved shape of East Lake Harbor led to an excessively long corridor in the East Lake–Yangchun Lake section compared to Corridor No. 27 in the research results. Excessively long and mislocated corridors can impede the transfer of ecological materials and energy between source places and weaken the ecological service capacity of ecological corridors. The existing zigzag corridors make it difficult to absorb and discharge stormwater and tend to cause stormwater accumulation, which is not conducive to the prevention of floods in urban areas.

Lastly, the lack of connecting corridors between the remaining source sites affects the flow of ecological materials and energy. Likewise, it does not facilitate the discharge and dissipation of stormwater during the flood season, weakening the resilience of the main urban area. 

### 5.2. Exploration of the Development Model

The largest number of ecological corridors was found in the Hanyang Ecological Zone, which is the product of *the Hanyang District Six Lakes Linkage Project* [59] ongoing since 2005 that was originally developed to curb the deterioration of lakes’ water quality with the natural water power of the Yangtze River. The construction method incorporated parallel blueways and greenways, that is, it included artificial river channels and supplemented them with greenways on either side. Firstly, the interconnected lakes can increase the overall storage capacity by increasing water storage space, thus offering an advantage in dealing with floods. Constructing greenways can also provide a path for stormwater transfer. Likewise, the flexible underlaying surface can increase the infiltration of stormwater during the transfer process and reduce the water amount. During the construction of ecological corridors, the lakes were seen as ecological sources able to curb the damage from urban construction activities. Corresponding plants in the areas were selected to control runoff pollution in response to the local runoff pollution characteristics of the study area.

The ESP that is adaptive to floods aims to build a resilient city and connect the major lakes in the main urban area of WUH through ecological corridors to form a regional and systematic ecological network. This network can, in turn, improve the ecological resilience of the main urban area. Through the flood-adaptive design of the ESP, the threshold of precipitation from floods is increased, while the speed of flood dissipation and transfer is enhanced, giving rise to a city resilient to floods.

### 5.3. Methodology Advantage

Compared to traditional gray infrastructure with a single function and no integrity, an ESP adaptive to floods is more holistic, diverse, economical, environmentally friendly, and less impactful.

Although traditional gray infrastructure is taller and more solid, it loses its single function [60] during non-flooding periods. The hard surface of the infrastructure is neither aesthetically pleasing nor safe, as well. The ESP that is adaptive to floods can not only cope with floods in urban areas but can also provide recreational and ecological services for urban residents during the non-flooding period.

Furthermore, traditional gray infrastructure negatively impacts the environment [61,62]. Hard levee separates lake and river ecosystems from other ecosystems, blocking the normal exchange of ecological materials and energy and increasing the risk of ecological fragility. The construction of traditional gray infrastructure, such as levees, not only occupies more of the lake’s natural space, but also deprives the lake of its hydrophilic vitality. However, the ESP that is adaptive to floods adopts a “sparse” approach to enhance ecosystem service capacity of rivers and lakes while also minimizing the impact of floods. In addition, the natural force of the river can be used to improve the water quality of the urban lakes by connecting them to natural water systems, such as the Yangtze and Han Rivers.

In terms of the construction impact, the construction of ecological corridors does not have to happen overnight. The corridors can be built first as key ones and then as potential ones. This step-by-step construction has less impact on the corridors’ functionality and can greatly reduce construction investment. This research developed two specific ways of construction: ① it adopted the culvert connection, i.e., maintaining the elevation of the roadbed and digging under the road to form the culvert. This specific method was used in road construction of Mudu Ancient Town in Wuzhong District, Suzhou. The advantage is that the construction cost is low, while the disadvantages are that the corridor cannot realize the functions of leisure and recreation, runoff pollution control, and cultural education, and the public perception is also greatly reduced; ② this research also used an artificial river with a road bridge above it, so that the part under the bridge becomes a park and the bridge becomes a road. This example refers to the effect of the Wuhan City Hanyang District–South Third Ring Road–Wetland Park–Meizi Interchange. However, its disadvantages are high construction costs, long lead time, and management difficulties.

In terms of economic impact, ESP has lower construction and maintenance costs and adds ecological and cultural value [63]. On the other hand, traditional gray infrastructure is not only costly, but also puts a certain amount of pressure on local finances for maintenance.

Compared to sponge cities, an ESP that is adaptive to floods is holistic and dynamic, complements the static defense of sponge cities, and relieves their storage pressure. Taking advantage of the natural hydraulics of the Yangtze and Hanjiang Rivers supplemented by plants can make up for the deficiencies of the sponge city in controlling runoff pollution [20]. An ESP that is adaptive to floods is also more advantageous in terms of cultural and educational services and public perception. The combination of a flood-adaptive ESP and sponge city can effectively improve a city’s resilience.

Based on the existing green space and river system in the main urban area of WUH, it was found that the existing blueways were limited in path selection by straight lines or original old river channels. The ESP that is adaptive to floods proposed in this research used the SCS-CN model to visualize and express the retention of surface water volume during floods. Because stormwater management should be given priority, corridor paths were extracted based on cost and distance using the MCR model. In traditional ESP studies [64,65], descriptions of the ecological corridor locations are not specific enough, causing difficulties in locating them precisely during construction. In this research, the specific locations of corridors were narrowed down to the street level. The suggestions about construction were made by comparing existing plans, so the research results are more practical, which is beneficial to governments for precise policy making and reducing decision-making costs.

## 6. Conclusions

This paper analyzed the spatial distribution of stormwater runoff under extreme precipitation and constructed an ESP that is adaptive to floods.

The SCS-CN model guided flood control objectives and derived the surface volume and spatial distribution of the runoff. The volume and spatial distribution were subsequently used to develop ecological corridors. The surface runoff path and the distribution of waterlogging points were used to optimize the corridor locations.

This research identified 19 ecological sources, 3 ecological zones, 32 ecological corridors, 10 key corridors, and 22 potential corridors using the gravity model. The Wuchang Ecological Zone exhibited a *three vertical and three horizontal* spatial distribution patterns with East Lake as the core. Furthermore, the Hankou Ecological Zone had a dispersed distribution pattern with Chestnut Lake–Huanzi Lake Group as the center. Lastly, the Hanyang Ecological Zone showed a clustered distribution. The research finally proposed a *two-axis and three-core* urban ecological resilience optimization strategy for decision makers, consisting of three ecological zones and two rivers.

In conclusion, enhancing urban ecological resilience can help cities cope with severe floods and can provide new methods and approaches for rainfall and flood control in Wuhan. Likewise, it can provide ideas for maintaining the ecological service capacity of lakes, realizing urban resilience, and ensuring regional ecological security.

The shortcomings of this research are as follows. (1) Hydrological heterogeneity within the same land type was not studied due to data limitations. Therefore, landscape design for controlling runoff pollution needs further research. (2) This research proposed the location of the corridors, but did not clarify their width, so subsequent studies should supplement this aspect to serve urban construction better. (3) Since there are few examples of how to construct flood-adaptive ESPs, it was difficult to quantify their functionality; thus, future studies need to address this problem. These questions are to be addressed in a follow-up study.

## Figures and Tables

**Figure 1 ijerph-20-00385-f001:**
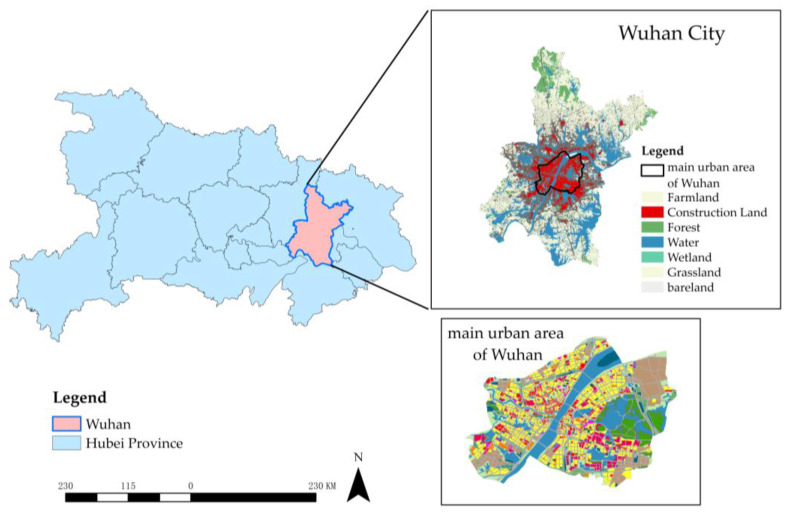
Map of the main urban area in Wuhan.

**Figure 2 ijerph-20-00385-f002:**
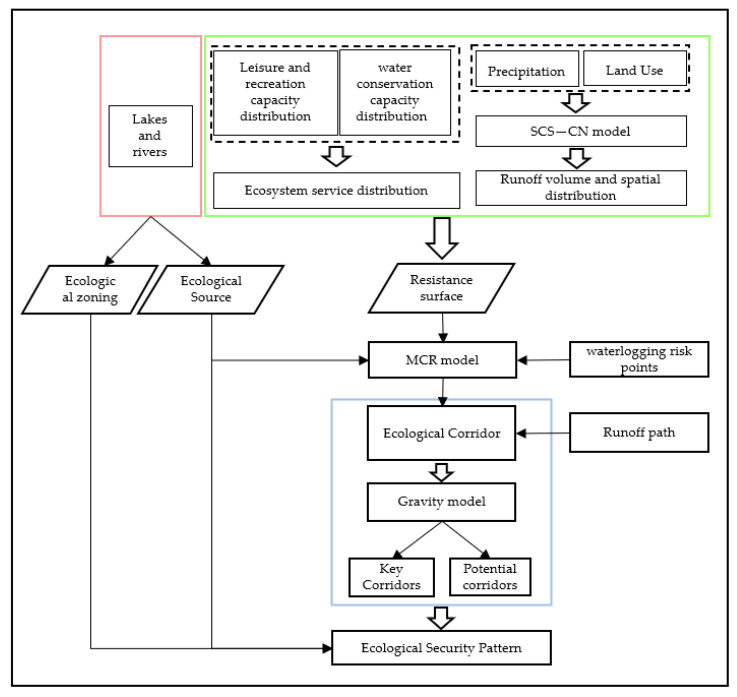
The methodological framework of the study.

**Figure 3 ijerph-20-00385-f003:**
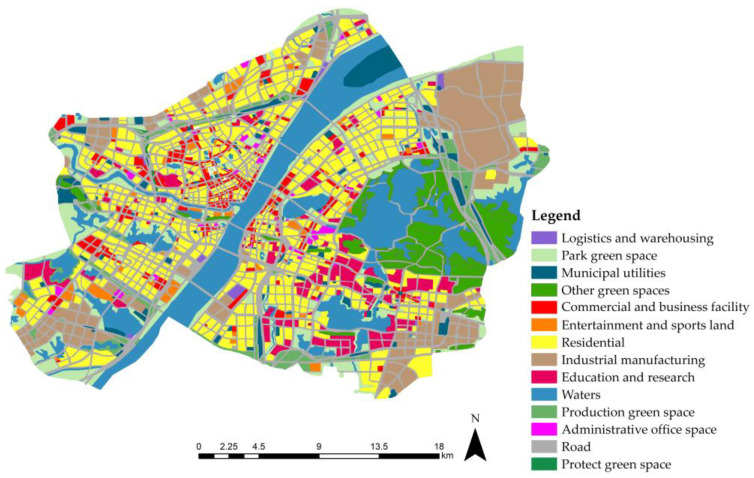
Distribution of land types on vector map.

**Figure 4 ijerph-20-00385-f004:**
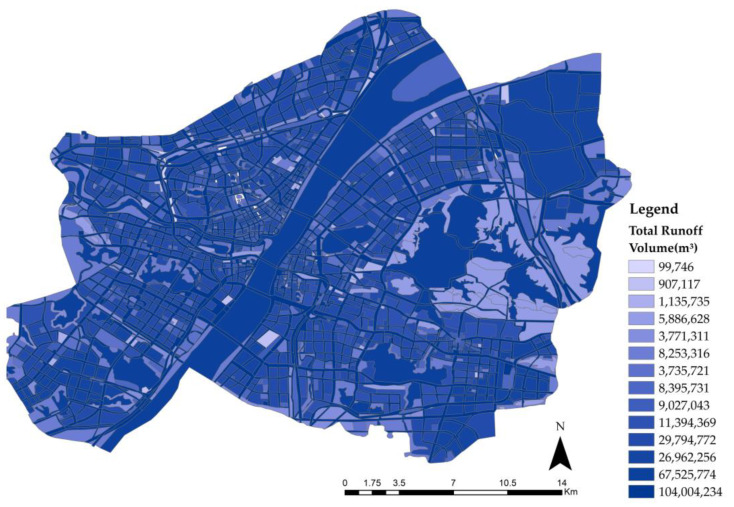
Spatial distribution of total runoff volume.

**Figure 5 ijerph-20-00385-f005:**
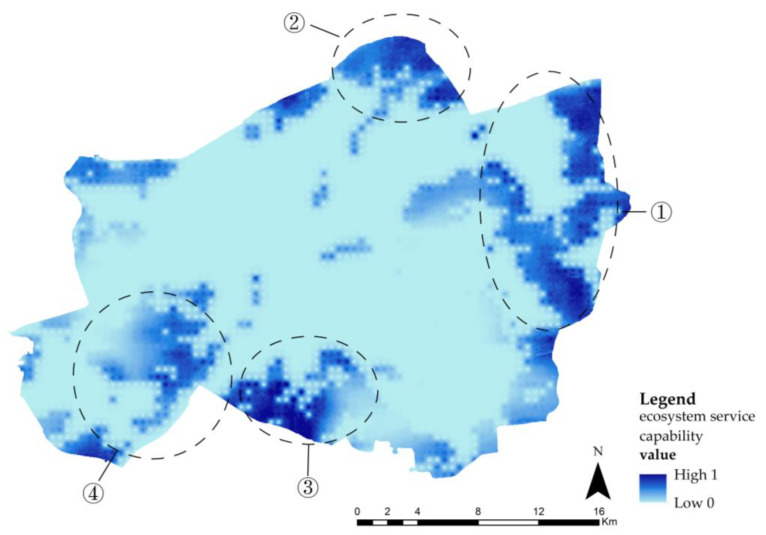
Distribution of the water conservation capacity.

**Figure 6 ijerph-20-00385-f006:**
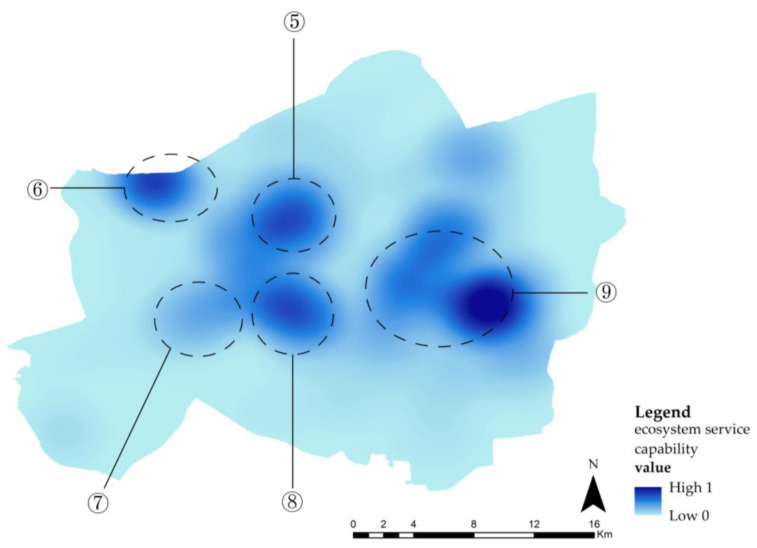
Distribution of the leisure and recreation capacity.

**Figure 7 ijerph-20-00385-f007:**
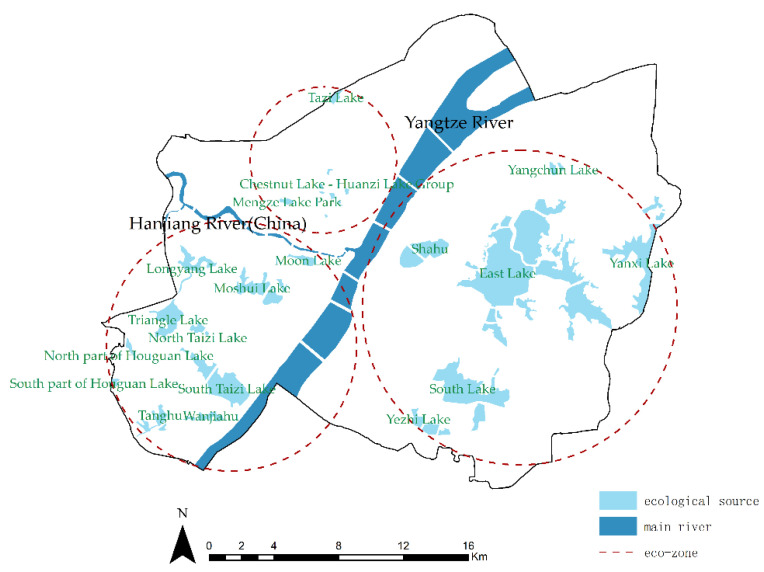
Distribution of ecological sources.

**Figure 8 ijerph-20-00385-f008:**
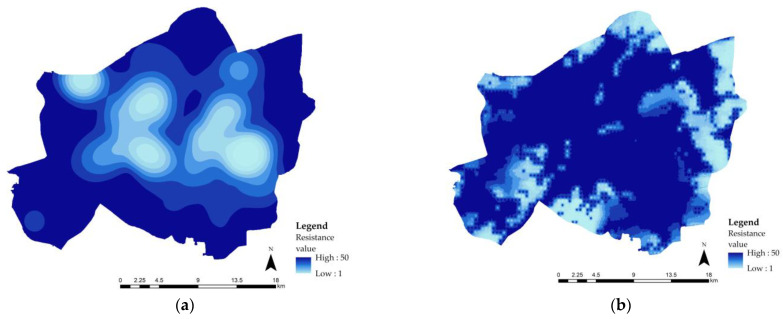
Resistance distribution: (**a**) leisure and Recreational resistance distribution, (**b**) Water conservation resistance distribution, (**c**) Runoff resistance distribution and (**d**) Comprehensive resistance distribution.

**Figure 9 ijerph-20-00385-f009:**
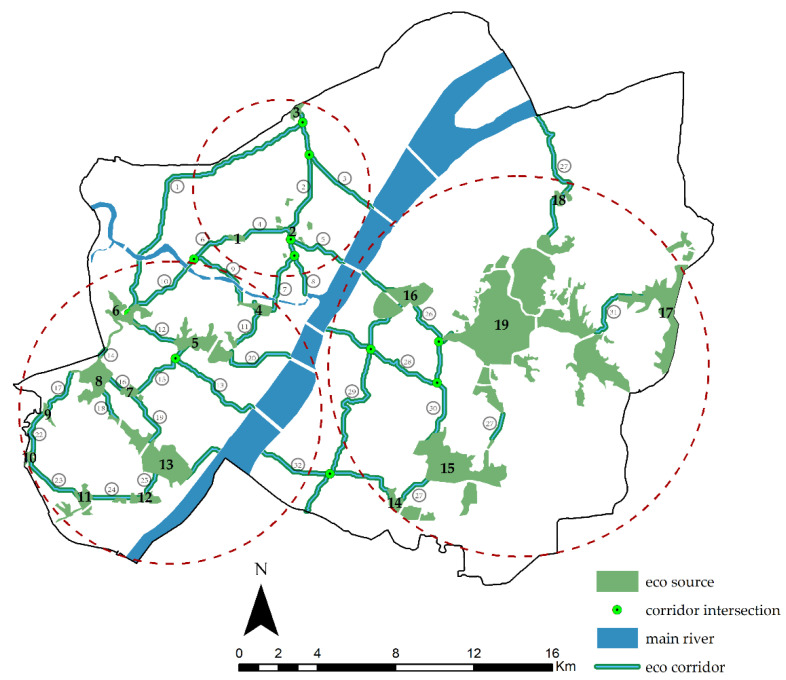
Distribution of ecological corridors.

**Figure 10 ijerph-20-00385-f010:**
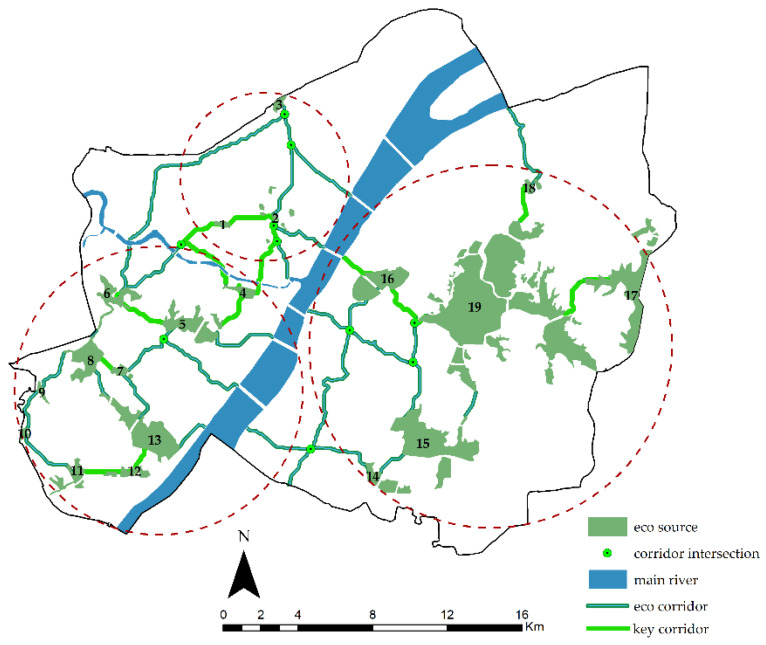
Distribution of key corridors.

**Table 1 ijerph-20-00385-t001:** SCS model soil hydrology group classification criteria.

Soil Hydrology Group	Soil Texture	Minimum Permeability (mm/h)
A	Thick sand, thick loess, and agglomerated silt	7.26–11.43
B	Thin loess, sandy loam	3.81–7.26
C	Clay loam, thin sandy loam, low organic matter or high clay soil	1.27–3.81
D	Soils that swell significantly after absorbing water, plastic soils, some saline soils	0–1.27

**Table 2 ijerph-20-00385-t002:** Total runoff volume corresponding to each land type.

Land Use Types	CN Value	S (mm)	Q (mm)	Area (m^2^)	Total Runoff Volume (m^3^)
Protect green space	72	98.78	196.55	507,492.75	99,745.64
Logistics and warehousing	91	25.12	431.80	2,100,792.94	907,117.32
Administrative office space	82	55.76	297.80	3,813,765.88	1,135,734.66
Other green spaces	67	125.10	157.98	37,262,863.73	5,886,628.39
Production green space	75	84.67	223.09	16,904,581.75	3,771,311.36
Park green space	66	130.85	151.00	54,656,304.71	8,253,315.63
Entertainment and sports land	90	28.22	414.07	9,021,995.04	3,735,721.10
Municipal utilities	90	28.22	414.07	20,276,203.44	8,395,730.74
Commercial and business facility	92	22.09	450.41	20,041,809.81	9,027,042.93
Education and research	91	25.12	431.80	26,388,219.08	11,394,369.34
Residential	77	75.87	242.46	122,887,264.91	29,794,771.57
Industrial manufacturing	91	25.12	431.80	62,441,887.42	26,962,256.36
Waters	98	5.18	584.91	115,447,026.19	67,525,774.34
Road	98	5.18	584.91	177,813,282.11	104,004,234.32

## Data Availability

Data is contained within the article or Appendix A.

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
