# Peer review of "Constructing a Flood-Adaptive Ecological Security Pattern from the Perspective of Ecological Resilience: A Case Study of the Main Urban Area in Wuhan"

_ijerph, 2022, doi:10.3390/ijerph20010385_

Round 1
Author Response
Dear reviewer1
Thanks for commenting on my paper during your busy schedule, here is my response. Please see the attachment

Reviewer 2 Report
The case study "Constructing a Flood Adaptive Ecological Security Pattern from the Perspective of Ecological Resilience: A Case Study of the Main Urban Area in Wuhan" touches an important aspect of urban flood management. The authors need to improve the following aspects before publication
Each abbreviation should be written in full at its appearance, for eg RMC and SCS-CN are not defined in abstract. abstract need to be more concise.
The rainfall runoff modelling through SCS-CN forms the major portion of the manuscript yet, no background literature was incorporated, at least authors should provide references to the Eq. 1-5. A lot of literature is available on this method authors can find following article useful
Credibility of design rainfall estimates for drainage infrastructures: extent of disregard in Nigeria and proposed framework for practice. Natural Hazards. Article DOI: 10.1007/s11069-021-04889-1
Impacts of land use and land cover changes on peakdischarge and flow volume in kakia and esamburmbur sub-catchments of narok town, kenya. Hydrology, 8(2), 82.
Estimating runoff using SCS curve number method. International Journal of Emerging Technology and Advanced Engineering, 8(5), 195-200.
Although the analysis is interesting but the entire study lacks incite into financial study, therefore, implementation of such projects will be difficult until financial aspect is explored, it would be better if the authors can analyze and mention financial aspect as well in the same article rather then writing a companion manuscript for financial aspect.
The text in Figure 5-8 are not readable so the authors are suggested to revise these figures.
Authors can present different ecological corridors in tabular form rather than text to improve the readability of manuscript (line 285-333)
Author Response
Dear reviewer2
Thanks for commenting on my paper during your busy schedule, here is my response. Please see the attachment

Reviewer 3 Report
The paper entitled “Constructing a Flood Adaptive Ecological Security Pattern from the Perspective of Ecological Resilience: A Case Study of the Main Urban Area in Wuhan” presents a comprehensive study of the ecological resilience and security related to flood. The authors developed a model which could be important not only for further scientific investigation but also applicable in the practice. The authors also presented the advantages and limitations of the proposed model. However, some corrections are still needed.
Taking into account that the Wuhan area is vulnerable to floods and other disasters, this territory was the subject of many other studies. Some of them applied similar methodology and should be also mentioned in this paper in the Introduction section (in previous investigations) and/or in the Discussion comparing with authors' results. I could recommend the following studies:
Fan, F., Wen, X., Feng, Z., Gao, Y., & Li, W. (2022). Optimizing urban ecological space based on the scenario of ecological security patterns: The case of central Wuhan, China. Applied Geography, 138, 102619.
Zhang, M., Peng, C., Shu, J., & Lin, Y. (2022). Territorial Resilience of Metropolitan Regions: A Conceptual Framework, Recognition Methodologies and Planning Response—A Case Study of Wuhan Metropolitan Region. International Journal of Environmental Research and Public Health, 19(4), 1914.
Jia, H., Liu, Z., Xu, C., Chen, Z., Zhang, X., Xia, J., & Shaw, L. Y. (2022). Adaptive pressure-driven multi-criteria spatial decision-making for a targeted placement of green and grey runoff control infrastructures. Water Research, 212, 118126.
Peng, Y., & Reilly, K. (2021). Using nature to reshape cities and live with water: an overview of the Chinese Sponge City programme and its implementation in Wuhan. European project, Grow Green.
Yang, J. J., & Zhu, X. (2017). Adapting urban water utilities to climate uncertainties: a case study of Wuhan, PRC. Procedia engineering, 198, 496-510.
Several references do not support the text and should be deleted and/or replaced with appropriate ones. Reference(s) should be provided for all data, equation(s), etc. which is not the author's own work.
The structure of the paper could be improved. Some parts (marked in the manuscript body) from the Results should belong Methodology section.
The quality of the text in Figure(s) should be improved.
English could be improved.
Other comments are given in the manuscript body.

Author Response
Dear reviewer3
Thanks for commenting on my paper during your busy schedule, here is my response. Please see the attachment

Round 2
Author Response
Dear Reviewer1
Thank you for reviewing my article in your busy schedule, here is my response "Please see the attachment."
